# Bioinformatic Analyses of Peripheral Blood Transcriptome Identify Altered Neutrophil-Related Pathway and Different Transcriptomic Profiles for Acute Pancreatitis in Patients with and without Chylomicronemia Syndrome

**DOI:** 10.3390/biom13020284

**Published:** 2023-02-02

**Authors:** Chia-Lun Liu, Yang-Hong Dai

**Affiliations:** 1Department of Internal Medicine, Tri-Service General Hospital, National Defense Medical Center, Taipei 105, Taiwan; 2Department of Radiation Oncology, Tri-Service General Hospital, National Defense Medical Center, Taipei 105, Taiwan

**Keywords:** acute pancreatitis, chylomicronemia syndrome, neutrophil degranulation, neutrophil, peripheral blood

## Abstract

Acute pancreatitis (AP) is a serious inflammatory condition of the pancreas that can be associated with chylomicronemia syndrome (CS). Currently, no study has explored the differences between non-CS-associated AP and CS-associated AP in terms of gene expression. Transcriptomic profiles of blood samples from patients with AP were retrieved from GSE194331 (non-CS-associated) and GSE149607 (CS-associated). GSE31568 was used to examine the linkage between non-CS-associated AP and the expression of micro RNAs (miRNAs). Differentially expressed genes (DEGs) were identified, a gene regulatory network was constructed, and hub genes were defined. Subsequently, single-sample gene set enrichment analysis (ssGSEA) scores of hub genes were calculated to represent their regulatory-level activity. A total of 1851 shared DEGs were identified between non-CS-associated and CS-associated AP. Neutrophils were significantly enriched in both conditions. In non-CS-associated AP, miRNAs including hsa-miR-21, hsa-miR-146a, and hsa-miR-106a demonstrated a lower expression level as compared with the healthy control. Furthermore, the expression patterns and regulatory activities were largely opposite between non-CS-associated and CS-associated AP, with significantly lower estimated neutrophils in the latter case. In summary, we found that the regulation of neutrophils was altered in AP. There was a different gene expression pattern and lower estimated neutrophil infiltration in CS-associated AP. Whether these findings are clinically significant requires further investigation.

## 1. Introduction

Acute pancreatitis (AP), which is often characterized by upper abdominal pain and elevated levels of pancreatic enzymes in peripheral blood, is an inflammatory condition of the pancreas. In the United States, AP is considered to be a major gastrointestinal disorder that necessitates hospitalization in most cases [1]. In terms of etiology, the leading causes of AP are gallstones and alcohol, which account for approximately two-thirds of cases [2,3]. Other causes include hypertriglyceridemia (HTG), post-endoscopic retrograde cholangiopancreatography, genetics, and certain medications [4,5,6,7]. Among them, HTG-induced AP (HTGP) is often underestimated, even though up to 35% of AP-related hospitalizations may be caused by HTG [8].

HTG can occur due to a number of conditions, ranging from rare familial disorders such as familial chylomicronemia syndrome (Familial CS, FCS) to acquired diseases like diabetes mellitus [9,10,11]. In fact, the spectrum of conditions associated with HTG may be linked to predisposing genetic risks [12]. In the case of FCS, reduced lipoprotein lipase (LPL) activity has been found to be caused by defects in LPL gene products or genes involved in the regulation of LPL activity, including APOC2, APOA5, GBIHBP1, and LMF1 [11]. In contrast, multifactorial chylomicronemia syndrome (MCS) occurs when an underlying genetic tendency is complicated by several medical factors, which in turn aggravates HTG [13]. For both types of CS, the most daunting complication is AP, and identifying patients with CS-associated AP is essential as genetic alterations may portend therapeutic resistance, especially in FCS [11]; and identification of genomic biomarkers for this special phenotype is important for long-term management and prevention of future episodes of AP [14]. However, no study has addressed the contribution of such biomarkers to the difference between CS- and non-CS-associated AP.

Currently, most studies investigating the genetic risk of AP are primarily based on the exploration of pancreatic tissues. Obtaining tissue samples, although risk stratification could be evaluated, may not be practical because the current diagnosis of AP is based on clinical findings, laboratory data, and imaging [15]. Nevertheless, with the progress of high-throughput technologies, such as DNA microarray and RNA-Seq, blood samples may still be of diagnostic value if these data are obtained [16]. In fact, multiple blood biomarkers have been found to be of diagnostic value for AP, such as lipase, amylase and trypsinogen [17]. These biomarkers are based on the biochemical profiling, and can sometimes be affected by other gastrointestinal diseases or disease conditions. Other indicators of AP such as those based on the metabolome, genes, cell free DNA (cfDNA) and miRNA, however, are different from the biochemical biomarkers and have not been widely investigated. These novel biomarkers are able to be integrated into machine-learning (ML) process, leading to accurate prediction of AP. Sun et al. used cfDNA methylation marker in the blood samples and constructed a prediction model for severe AP with high model performance and accuracy [18]. Zhang et al., explored the transcriptomic profiles of peripheral blood cells in AP [19]. Through ML methods, they found that S100A6, S100A9, and S100A12 were predictors of severe AP. By examining the perturbation of genomic/transcriptomic profiles in the blood samples, we can sufficiently characterize the altered biological pathways in a specific condition like CS.

In this study, we aimed to compare the transcriptomic profiles in the blood between non-CS-associated and CS-associated AP in an attempt to identify biomarkers that could discriminate between these two patient populations. We used peripheral blood samples and examined whether the associated transcriptomic profiles and biological pathways were different. Finally, we tried to construct an ML-based prediction model for AP in both conditions.

## 2. Materials and Methods

### 2.1. Samples

The series matrix files GSE194331, GSE149607, and GSE31568 were downloaded from the National Center for Biotechnology Information Gene Expression Omnibus (GEO; http://www.ncbi.nlm.nih.gov/geo, accessed on 11 September 2022) using the R/Bioconductor GEOquery package. The datasets were derived from whole-blood samples [20,21,22,23]. The GSE194331 dataset was based on RNA-Seq (platform: Illumina HiSeq 2500) and included 87 patients with AP and 32 healthy controls (total = 119). In addition, 87 patients were annotated with severity according to the revised Atlanta classification. Raw RNA-seq data were extracted for differential expression analysis and normalized using variance-stabilizing transformation for cross-sample comparison using the R/Bioconductor DESeq2 package [24]. The expression of genes among different severities was drawn in a uniform manifold approximation and projection (UMAP) space to visualize the data distribution using the umap package in R [25]. The GSE149607 dataset was based on a DNA microarray (platform: Affymetrix Human Gene 2.0 ST Array) and contained 47 patients with CS (FCS:19; MCS:28) and 15 healthy controls (total = 61). Data from GSE149607 were normalized using the robust multi-array average (RMA) according to the authors’ methods. GSE31568 was based on non-coding RNA profiling (febit Homo sapiens miRBase 13.0) and was used to identify potential micro RNAs (miRNAs) associated with AP. This study included 38 patients with AP and 38 healthy controls (76 patients). This dataset was normalized using median and quantile normalization.

### 2.2. Differential Expression Analysis

For GSE194331, differentially expressed genes (DEGs) were identified in patients with AP with varying severities and healthy controls through a linear regression model using the R/Bioconductor limma package [26]. Significance was defined as a false discovery rate (FDR)-adjusted p-value (adj. p) < 0.05. A Venn diagram (https://bioinformatics.psb.ugent.be/webtools/Venn/, accessed on 11 September 2022) was created to visualize the common overlap of DEGs among the three severities (mild AP vs. control; moderately severe AP vs. control; severe AP vs. control). A volcano plot was generated to visualize the overall distribution of DEGs for any severity of AP vs. control using the EnhancedVolcano package in R. Similarly, for CS cases in GSE149607, DEGs were identified in AP subjects and healthy controls and were then intersected with the common overlapping DEGs among the three severities in GSE194331 in order to filter out genes for over-representation analysis (ORA).

### 2.3. Gene Set Enrichment Analysis (GSEA)

A ranked list was generated according to the log fold change (logFC) for DEGs between all AP cases and healthy controls. GSEA was performed using the GSEA function of the R/Bioconductor clusterProfiler package [27]. The enrichment score (ES) was derived by calculating the weighted Kolmogorov–Smirnov statistic to a running sum of the ranked list. The ES was further normalized to account for the size of each gene set. FDR’s less than 0.05 were considered statistically significant.

### 2.4. Over-Representation Analysis (ORA)

To identify the pathways involved in AP, common DEGs between GSE194331 and GSE149607 were inputted into Metascape (https://metascape.org/, accessed on 11 September 2022), a website designed to provide comprehensive gene list annotation and pathway enrichment analysis [28]. During enrichment analysis, the input DEGs were compared to thousands of gene sets curated from various sources (KEGG Pathway, GO Biological Processes, Reactome Gene Sets, Canonical Pathways, CORUM, and WikiPathways) to identify their involvement in specific biological processes. Enriched terms were defined by hypergeometric test and Benjamini-Hochberg *p* correction algorithm. *p* < 0.05 was considered significant [28].

### 2.5. Construction of Regulatory Gene Network and Identification of Hub Genes

A protein–protein interaction (PPI) network was constructed by applying DEGs in the Search Tool for the Retrieval of Interacting Genes (STRING) database. The network was exported and reconstructed using Cytoscape software (version 3.8.2). Sub-network modules were identified using plug-in molecular complex detection (MCODE). The criteria for determining complexes of biological significance for AP were an MCODE score > 3 and a node number > 5. Genes with at least ten interactions were considered hub genes. CytoHubba was used to identify the central elements of the biological networks [29]. The top 10 nodes from five methods, namely, Maximum neighborhood component (MNC), maximal clique centrality (MCC), edge percolated component (EPC), the density of maximum neighborhood component (DMNC), and degree, were obtained, and genes with degrees less than 10 were eliminated.

### 2.6. Identification of Target miRNAs

It has been shown that altered miRNA expression may cause the alteration of physiological functions involved in inflammation, including AP [30]. To identify the miRNAs that regulated the hub genes for AP in both datasets (GSE194331 and GSE149607), miRTargetLink 2.0 (https://www.ccb.uni-saarland.de/mirtargetlink2, accessed on 11 September 2022) was used. A unidirectional search was conducted to identify the target miRNAs using hub genes as inputs. Only target miRNAs that were strongly validated were retained. A literature search was then performed to confirm the association between the target miRNAs and the biological mechanisms of interest. Normalized intensity values from GSE31568 were used to validate target miRNAs.

### 2.7. Single Sample Gene Set Enrichment Analysis (ssGSEA)

To estimate the enrichment of hub genes under AP and health status in patients with or without CS, the single-sample GSEA (ssGSEA) score was derived using the ssGSEA function of the R/Bioconductor GSVA package [31]. The ssGSEA score represents the degree to which the hub genes are coordinately up- or down-regulated for each sample [32]. The ssGSEA score of each sample was then min–max scaled.

### 2.8. Immune Cell Infiltration Analysis

Immune cell type enrichment analysis from gene expression data was conducted to estimate immune cell infiltration for each sample in GSE194331 and GSE149607. ssGSEA scores for immune cell-specific gene sets integrated using publicly available methods were calculated using the ConsensusTME package in R [33]. Four studies aiming to estimate immune cell infiltration were adopted, including xCell, MCP-counter, Bindea et al., and Danaher et al. [34,35,36,37].

### 2.9. Prediction of Associated Genes and Biological Pathways

We hypothesized that the expression profiles of hub genes would be different between patients with and without CS. To confirm this, we first calculated Spearman’s correlation coefficients between hub genes and specific immune cell types of interest. The correlation coefficients were then averaged across four studies (xCell, MCP-counter, Bindea et al., and Danaher et al.) and ranked. We then extracted the top ten genes with positive correlation coefficients and used GeneMANIA to construct new PPI networks [38]. GeneMANIA is an online server where interconnections between proteins can be explored in terms of physical interactions, co-expression, prediction, and co-localization. Pathway analysis based on the DAVID website was performed to identify the functional enrichment of the top ten genes and interacting genes. Enrichment is measured by the hypergeometric *p*-value, and *p* < 0.05 was considered significant.

### 2.10. Construction of Diagnostic Models for AP

Patients with AP in GSE194331 and GSE149607 datasets were randomly divided into training and testing sets based on an 8:2 ratio. The least absolute shrinkage and selection operator (LASSO) regression algorithm was used to select 43 hub genes with non-zero coefficients via 10-fold cross-validation [39]. LASSO models were built based on the selected gene signatures. Receiver operating characteristic (ROC) analysis with the area under the curve (AUC) and a 95% confidence interval (CI) was conducted to evaluate the performance of the LASSO models using the pROC package in R in the training, testing, and entire sets [40].

### 2.11. Statistical Analysis

All statistical analyses were conducted using the R software (version 4.1.2). Wilcoxon’s rank sum test was performed for continuous variables under specific conditions, and statistical significance was set at *p* < 0.05.

## 3. Results

### 3.1. Characterization of Transcriptomic Profiles in AP without CS

Transcriptomic data from the peripheral blood of AP patients without CS with varying severities were examined using UMAP analysis. Except for patients with mild AP, who were mixed with healthy controls, patients with moderate and severe AP were different and clustered together (Figure 1A). Differential gene analysis was performed separately, and Venn diagram analysis identified an overlap of 6513 genes among the different severities of AP (Figure 1B). The most prominent DEGs identified in the AP samples were ARG1, S100A8, S100A12, and ANXA3 (Figure 1C). GSEA revealed an enriched Gene Ontology (GO) term:0042581: specific granule (normalized ES:2.47, adjP:0.0319). A gene-concept network was constructed, and core enriched genes were identified: ARG1, CD177, MCEMP1, ANXA3, ORM1, ORM2, HP, SLPI, CLEC4D, RETN, and GPR84 (Figure 1E). Specific granules are a type of secretory vesicle found in granulocytes [41]. It is specifically applied to neutrophils, which release parts of their granule contents in response to lipopolysaccharide and proinflammatory proteins, such as S100A9 [42,43]. Some typical contents of the specific granules are shown in Figure 1F. Compared with healthy controls, expression levels of the enriched core genes were higher in patients with AP (Figure 1G). In GSE149607, there was clustering of control samples that was separate from FCS and MCS (Appendix A). The most prominent DEGs were identified in the control samples, which were MS4A2, CPA3 and LRG1 (Appendix A). GSEA results showed 3819 enriched GO terms and GO:0042581 was also significantly enriched, but with the opposite regulatory pattern as compared with that in GSE194331 (Normalized ES:–2.68, adj. p < 0.001, Appendix A). Based on the findings derived from the blood samples, these results suggest that neutrophil-related biological processes play vital roles in AP and the regulatory activity might be different between CS-associated and non-CS-associated AP.

### 3.2. Neutrophil Degranulation Pathway Is Altered in AP with and without CS

Patients with CS are at higher risk of developing AP. Patients from GSE149607 were evaluated in order to investigate whether they shared similar altered biological pathways with GSE194331. We found a substantial overlap of DEGs (n = 1851) between these two patient populations (Figure 2A). For GSE149607, over 50% of the DEGs (1851/3510, 52.7%) were shared with GSE194331. Next, the shared DEGs were inputted into Metascape for ORA. Surprisingly, neutrophil degranulation (Reactome: R-HSA-6798695) was enriched as the top GO term (Figure 2B). To identify the hub genes for AP in both conditions, 1851 common DEGs were used to construct a gene regulatory network using Cytoscape (Figure 2C). Through sub-network analysis using CytoHubba, we found that STAT3, IL1B, TLR4, MYC, PTPRC, ITGAM, MAPK3, and ACTB were selected three times among the five algorithms (Appendix A). Despite the high correlation between the two platforms regarding DEGs, their correlation was negative (–0.7, *p* = 4.9 × 10^−7^) (Figure 2D). In support of this, heatmaps of the up-regulated and down-regulated hub genes for CS- and non-CS-associated AP demonstrated strikingly different expression profiles (Figure 3A,B). Overall, these findings suggest that AP patients with CS shared the same biological alterations as those not diagnosed with CS; however, the transcriptomic profiles may be different.

### 3.3. Identification of miRNAs

The hub genes were input into miRTargetLink 2.0 to build an interaction graph. As shown in Figure 3C, a central component of hub genes with high node degrees was identified, including SOCS3, IL6R, CD274, STAT3, TLR2, and TLR4. The miRNAs with more than two interactions with the central hub genes were hsa-miR-124-3p, hsa-miR-21-5p, hsa-miR-146a-5p, hsa-miR-23a-3p, hsa-miR-106a-5p, and hsa-miR-125b-5p. Using plasma samples of AP and healthy controls from GSE31568, we found that the expression levels of hsa-miR-21, hsa-miR-146a, and hsa-miR-106a were significantly lower than those in the controls (Figure 3D). A literature search revealed that these miRNAs in peripheral neutrophils were also down-regulated under various inflammatory conditions (Table 1) [44,45,46,47,48,49]. Therefore, as neutrophils account for two-thirds of white blood cells, plasma hsa-miR-21, hsa-miR-146a, and hsa-miR-106a could be used as potential biomarkers for AP.

### 3.4. Regulatory Activity of the Hub Genes Was Different in Patients with CS

The ssGSEA scores of the hub genes were calculated, and we found that the scores were significantly different between the AP and control samples in both datasets (all *p* < 0.001) (Figure 4A). However, in GSE149607, the ssGSEA scores were lower in patients with CS (FCS and MCS) than in the controls. Furthermore, ssGSEA scores demonstrated a high correlation with neutrophil infiltration estimated via signatures derived from xCell, Bindea et al., Danaher et al., and MCP-counter (Figure 4B). These combined findings suggest that the regulatory-level activity of hub genes in AP patients was dysregulated regardless of the presence of CS. However, there were opposite profiles of neutrophil infiltration in the peripheral blood.

### 3.5. Correlation Profiles of Hub Genes in Patients with CS- and Non-CS-Associated AP

To investigate which hub genes were responsible for the differences observed between CS- and non-CS-associated AP, we correlated the expression levels of the hub genes with regulatory activity for AP (i.e., ssGSEA scores). The correlation profiles were similar, especially for the down-regulated and up-regulated genes in GSE194331 and GSE149607, respectively (Figure 5A,B). However, the top 10 hub genes for the other gene sets with the opposite regulation were slightly different. In GSE194331, the top 10 genes were TLR5, SPI1, SOCS3, SELL, NCF4, LY96, ITGAM, IL1R1, IL1B, FCGR1A, TNFRSF1A, TLR6, TLR4, SPI1, NCF4, NCF2, IL1R1, IL1B, FCGR3B, and CXCR2 in GSE149607.

By applying GeneMANIA to each dataset, networks of the top ten genes and genes with interactions, either predicted or not, were constructed (Figure 6A,B). The genes identified by this data source were different and demonstrated varying linking profiles. The only shared genes were IL1RN, TICAM2, GATA1, IL1RAP, HOXA10, and NCF1. Functional enrichment analysis revealed differentially enriched GO terms. In GSE194331, the enriched terms were growth factor receptor binding, the cellular response to lipopolysaccharide, and the cellular response to interleukin-1, whereas in GSE149607, enriched terms included the cellular response to molecules of bacterial origin, the cellular response to biotic stimulus, and the response to molecules of bacterial origin. Because the expression of the top 10 genes in GSE149607 was lower in subjects with AP than in the controls, it may be likely that the activity of the functional pathways is suppressed or absent.

### 3.6. Diagnostic Models for CS- and Non-CS-Associated AP Based on Hub Genes

To identify candidate genes with diagnostic value for AP, we selected potential genes with predictive ability from the 43 hub genes. The parameters were tuned, and genes with non-zero coefficients were identified (Figure 7A,B). Fifteen genes were selected from the GSE194331 dataset (TLR8, STAT3, IL7R, TNFRSF1A, TLR3, NLRP3, CD83, FOXP3, FAS, TLR6, LY96, ITGAX, FCGR1A, NCF2, and IL1R1). Based on these genes, a diagnostic model was constructed, and the predictive values were high, with AUC = 1 (95% CI:1-1), 0.925 (95% CI:0.8193-1), and 0.998 (95% CI:0.9941-1) in the training, testing, and entire sets, respectively (Figure 7C). Similarly, a diagnostic model for CS-associated AP (GSE149607) showed high performance based on the four selected genes (ICAM1, TNFRSF1A, CD83, and CXCR5), with AUC values of 0.996 (95% CI:0.9852-1), 0.889 (95% CI:0.6711-1), and 0.993 (95% CI:0.9797-1).

## 4. Discussion

Through integrated bioinformatic analyses of blood samples from subjects with AP, we identified 43 common hub genes in patients with and without CS. Both patient populations share the same pathways related to neutrophil degranulation. We found that the regulation of hub genes was essentially the opposite and could potentially aid in the discrimination between CS and non-CS causes of AP. The regulatory activity of the hub genes, characterized by ssGSEA scores, displayed significant differences between the AP and control samples. Compared with healthy controls, patients without CS had significantly higher ssGSEA scores, whereas patients with either FCS or MCS had significantly lower ssGSEA scores. Additionally, this activity score correlated well with the estimated neutrophil infiltration in the blood for both of the datasets. The correlation profiles in our study also showed similar patterns with slight differences. Furthermore, the selected hub genes in both datasets revealed high diagnostic values for AP. Therefore, these 43 hub genes could serve as potential biomarkers for AP, regardless of the presence of CS.

Because of the relative inaccessibility of pancreatic tissue and the rapid course of AP, obtaining blood samples seems advantageous for establishing early diagnosis [50]. At present, there is no gold standard laboratory test for diagnosing AP, and serum lipase activity at least three times greater than the upper limit is often adopted to assess pancreatic inflammation [51]. Other potential serum biomarkers include pancreatic isoamylase, pancreatic elastase, serum trypsin, urinary trypsinogen-activated peptide, phospholipase A2, and carboxypeptidase B [52,53]. However, these biomarkers have not been incorporated into clinical use for various reasons, such as their low diagnostic accuracy and availability [54].

AP is a type of inflammatory disease that begins with local inflammation of the pancreas and progresses into a generalized inflammatory response followed by multi-organ dysfunction [55,56,57,58]. The most widely accepted initiating event is the premature activation of trypsin, which activates a zymogen cascade that induces the attraction of leukocytes to the pancreas [17,59]. Therefore, the immune system is activated at the onset of AP in the bloodstream. However, the human immune system is complex and includes various types of immune cells. An overview of the activated immune response to infection or inflammation is impossible if we rely solely on certain molecules. Due to the advancement of high-throughput technologies such as DNA microarray, RNA-Seq, and NanoString, RNA abundance can be measured on large scales, and the complex immune system can be profiled in an unbiased manner [16]. To profile the human immune system, blood transcriptomic profiling has proven powerful in elucidating the course and pathogenesis of autoimmune diseases, infectious diseases, and cancer [60,61,62]. For example, previous studies have found that there were profound changes in transcript abundance in patients with local and systemic infections [63,64]. These changes have diagnostic potential and help evaluate the severity of infection during the course of the disease. Few studies have investigated the transcriptional profiles of AP in the blood. The most recent study was conducted by Zhang et al., who explored the transcriptome of peripheral blood cells in AP using different severity classifications [19]. They found that S100A6, S100A9, and S100A12 were predictors of severe AP, and that a specific subtype of neutrophils was responsible for COVID-19-induced AP. In their study, they identified that “neutrophil degranulation” was the central pathway in both patients with AP and COVID-19. They concluded that their findings could assist in the severity evaluation and research of AP-related conditions. Therefore, by exploring the transcriptome in blood, the pathogenesis of a disease can be better defined, and reliable biomarkers can be identified.

In AP, neutrophils are the first-responder leukocytes recruited to the injured site and play a vital role in pathogenesis during the early phase [65,66]. Additionally, it has been shown that there is abnormal signaling in peripheral blood neutrophils in AP complicated by organ dysfunction, suggesting the systemic effect of neutrophil activation [67]. Once activated and/or primed by mediators released by systemic inflammation, neutrophil degranulation may occur, releasing toxic enzymes and products that can damage the endothelium [68]. In our study, we found that “neutrophil degranulation” was significantly enriched in both CS- and non-CS-associated AP, indicating that this pathway was dysregulated under both conditions. Furthermore, when exploring their common hub genes, we identified STAT3 as one of the central regulatory elements of many sub-networks. This finding supports the key role of STAT3 in neutrophil functions, such as maturation, mobilization, and migration [69,70]. Moreover, STAT3 has been linked to many miRNAs, and we found that the expression levels of hsa-miR-21, hsa-miR-146a, and hsa-miR-106a were significantly lower in AP. Some evidence supports the findings of this study. For example, hsa-miR-21 suppresses the expression of STAT3 and is down-regulated in juvenile idiopathic arthritis [71]. The down-regulation of blood hsa-miR-146a has been associated with coronary artery disease, acute exacerbation of chronic obstructive pulmonary disease, and severe systemic lupus erythematosus [72,73,74], and down-regulated hsa-miR-106a has been found in the peripheral blood mononuclear cells of patients with chronic hepatitis B patients [75]. These results indicate that the down-regulation of these miRNAs could occur under specific inflammatory conditions. As AP is systemic inflammation, we believe that these miRNAs might be potential blood biomarkers for AP.

In our study, we found strikingly different trends in neutrophil regulation and the regulatory activity of hub genes in CS. This discrepancy might be caused by the accumulation of chylomicrons and triglycerides (TG) in the blood due to the accumulation of fat in the body. It has been shown that circulating levels of TG and remnants of chylomicrons positively correlate with neutrophils [76,77]. Therefore, baseline neutrophil infiltration in the bloodstream might be higher in CS patients. These results imply that neutrophil levels were higher in the CS group in the absence of AP. Additionally, the estimated decrease in neutrophils in patients with CS-associated AP may be the result of neutrophil migration to the sites of injury. Zhang et al. induced AP in mouse models and found that GBIHBP1 knockout mice tended to have more neutrophil infiltration in pancreatic tissues and were prone to developing large areas of pancreatic necrosis [78]. In addition, loss-of-function mutations in GPIHBP1 are one of the main genetic causes of FCS, thus patients with CS are more likely to have pancreatic necrosis and organ failure [6,79,80]. Based on current evidence, we hypothesized that neutrophils might be directed toward the injured site and sequestered in the inflamed pancreas. Furthermore, we found that CXCR2 was down-regulated in CS-associated AP. This gene encodes CXCR2, which is a major chemokine receptor involved in neutrophil trafficking [81]. When the expression of CXCR2 on the neutrophil membrane decreases, neutrophil migration is impaired, leading to the accumulation of neutrophils in the vessel wall [82]. However, this phenomenon was not observed in our study, where CXCR2 down-regulation was not correlated with an increase in estimated neutrophil infiltration. One likely explanation is that the neutrophils producing CXCR2 transcripts migrate to the pancreas, which indirectly suggests that neutrophils were largely mobilized in the presence of AP in patients with high baseline neutrophil levels. In support of this, the correlation profiles were similar despite different expression patterns, indicating that neutrophils shared similar activation/response to AP regardless of whether they were CS-associated or not. This indicates that neutrophils were more likely to be directed away from the blood as opposed to down-regulating genes. If this hypothesis is true, inflammation of the pancreas might be more severe, and aggressive management could be initiated early to prevent organ failure. This conclusion needs to be addressed carefully and validated through further experiments. Nevertheless, these findings provide insight into the systemic immune response in CS when AP occurs.

In this study, we examined each dataset separately. As there were two platforms for obtaining the transcriptomic data, the data cannot be merged directly because of the different scales of gene expression data generated. Additionally, the techniques for quantifying their expression are not the same. Since we attempted to identify common altered genes/pathways between two patient populations, we analyzed the data independently and pooled the results together. For example, we conducted DEG analyses separately and checked whether there was a strong correlation between the two datasets in terms of hub gene expression. If the correlation is strong, it means we can derive comparable results from the two datasets as the transcriptomic perturbation follows a trend for a specific condition, such as AP in our study. The strong Spearman’s correlation in our study indicates that the expression of hub genes in both datasets can be compared [83]. In terms of different patient population characteristics and their effect on our analyses, the differences in this study arise from the fact that AP is attributed differently in GSE194331 and GSE149607. In GSE194331, patients with AP do not have underlying CS, whereas patients in GSE149607 had a genetic predisposition toward developing AP throughout the patients’ lives. This is similar to studies that compare phenotypes between wild types and mutants. The purpose of such studies is to address the change in phenotypes due to shifts in genotypes and their relationships [84]. However, data imbalance can sometimes be an issue. In the case of a large genomic analysis such as GWAS, case-control ratios are often unbalanced (case:control = 1:10) or extremely unbalanced (case:control < 1:100) when the prevalence of a condition or a disease is low [85]. This can pose a tremendous challenge and potentially increase the type I error rate during the association study. However, in our study, even though the datasets were imbalanced, the case-control ratios are 1:0.37 in GSE194331 and 1:0.32 in GSE149607, respectively. Moreover, in our study, the datasets were primarily used for differential gene expression analysis. By using DESeq2, the average gene expression is compared to derive the fold change. Thus, a slight case-control imbalance might not affect the DEGs. However, according to the designer’s recommendation (https://support.bioconductor.org/p/9142704/, accessed on 11 September 2022), there is nothing to do if there is an extreme imbalance. Nevertheless, this situation is not present in our study. Additionally, the prediction model, without data augmentation and balancing, showed high AUC in training and testing datasets, suggesting the presence of a case-control imbalance did not dent the model performance. However, this should still be validated by external independent datasets.

Our study had several limitations. First, we analyzed only three datasets. Therefore, the power to obtain conclusive results may be limited. Second, there was a lack of data from the pancreatic tissue of the same study population. Therefore, we could not evaluate the difference in neutrophil infiltration between blood and tissue samples. Third, information on the TG levels in patients with CS is unknown. Thus, the relationship between TG level and estimated neutrophil infiltration could not be elucidated. Due to the rarity of FCS and MCS, we were unable to find suitable patients in our hospital to confirm the results. Despite these limitations, our study may provide insights into HTGP and other relevant disorders. Furthermore, as we derived the results only from patients with AP, other inflammatory conditions may have similar gene expression profiles when neutrophil plays a dominant role. Other clinical features related to diagnosing AP should be incorporated to improve the diagnostic yield. Lastly, the time for generating results from DNA microarray or RNA-Seq might be approximately one to two days [80]. It could be lengthy when immediate management of AP is necessary. We should still follow the current management guideline to prevent mortality, while at the same time wait for the results that address the patient’s underlying causes of AP. This approach could help clinicians spend time grappling with the pathogenesis and managing this disease in a long-term.

In conclusion, we performed integrated bioinformatic analyses and identified that the neutrophil-related pathway was altered in patients with AP. Neutrophil dysregulation occurs in both CS- and non-CS-associated AP. Diagnostic models based on the selected hub genes for each patient population displayed a high performance. By calculating the ssGSEA score of the hub genes and estimating the neutrophil infiltration in the blood, we found that patients with CS-associated AP had a significantly lower regulatory activity of the hub genes and correspondingly lower neutrophil infiltration, which implied possible mobilization of neutrophils to the injured pancreas. Our findings offer insight into the pathogenesis and immune system alterations in patients with CS-associated AP, facilitating further research on this syndrome.

## Figures and Tables

**Figure 1 biomolecules-13-00284-f001:**
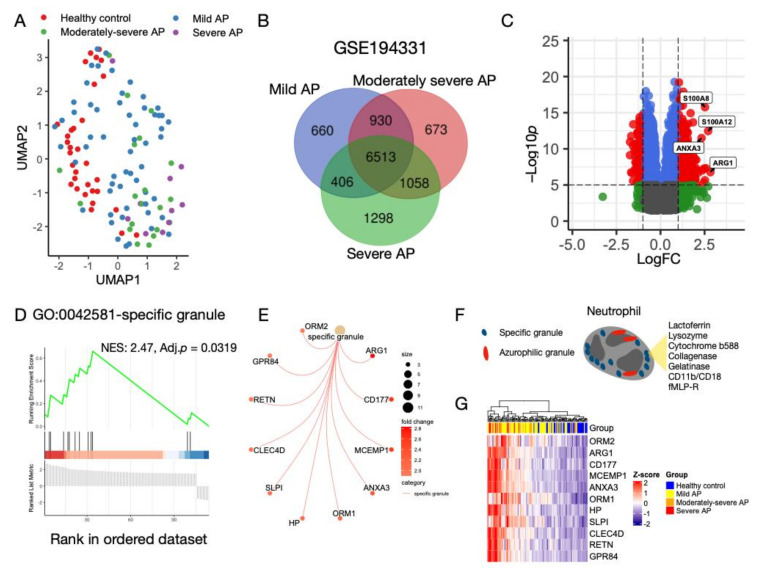
Characterization of gene expression profile in patients with non-chylomicronemia syndrome (CS)-associated acute pancreatitis (AP). (**A**) Uniform manifold approximation and projection (UMAP) plot of gene expression between AP samples and healthy controls in GSE194331. Patients with AP are divided into mild, moderately severe, and severe. Red: Healthy control. Blue: Mild. Green: Moderately severe. Purple: Severe. (**B**) Venn diagram of differentially expressed genes (DEGs) among three degrees of severity in GSE194331. There are 6513 common DEGs. (**C**) Volcano plot of the common DEGs in the Venn diagram. Red: Significantly up-regulated and down-regulated genes based on the |logFC| > 1.5 and the adj. *p*-value < 10 × 10^−5^. Blue: DEGs with |logFC| ≤ 1.5 and adj. *p* ≤ 10 × 10^−5^. Green: DEGs with |logFC| >1.5 and adj. *p* > 10 × 10^−5^. Gray: DEGs with |logFC| ≤ 1.5 and adj. *p* > 10 × 10^−5^. (**D**) Gene set enrichment analysis (GSEA) plot of gene ontology (GO): 0042581-specific granule. NES and adj. p are shown. (**E**) Gene-concept network of genes listed in GO: 0042581. Dot size of the GO term represents the number of DEGs that are annotated based on the term. Color bar indicates the fold change between AP and control. (**F**) Diagram of neutrophil and its common granules. Red: Azurophilic granule. Blue: Specific granule. Contents inside the specific granule are shown. (**G**) Hierarchical clustering heatmap shows expression levels of the annotated DEGs in GO: 0042581 in patients with varying severities and healthy subjects. Expression levels are z-transformed.

**Figure 2 biomolecules-13-00284-f002:**
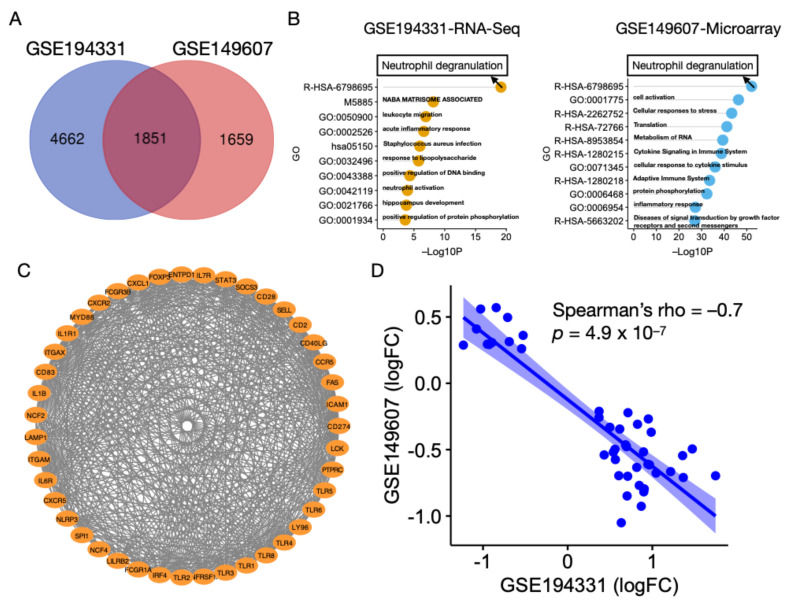
Common pathway in CS- and non-CS-associated AP. (**A**) Venn diagram of DEGs between AP and control in GSE194331 and GSE149607. (**B**) Dot plots show clustered enrichment ontology categories from over-representation analysis (ORA) in GSE194331 and GSE149607. –Log10-transformed multiple testing-adj. p-value is shown for each enriched term. Black arrows indicate the name of the top enriched term. (**C**) Hub genes obtained using MCODE in Cytoscape. (**D**) Scatter plot of correlation between logFC of common DEGs in GSE194331 and GSE149607. Significance of correlation was obtained using Spearman’s test.

**Figure 3 biomolecules-13-00284-f003:**
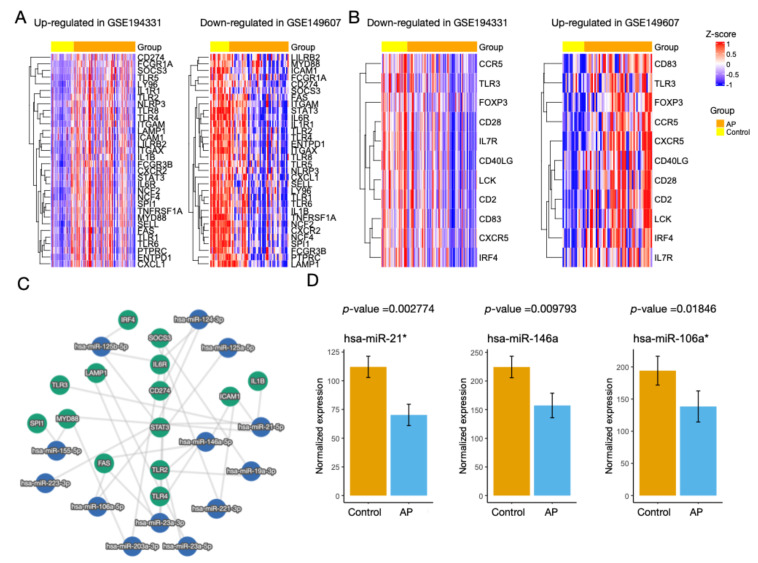
Regulation of hub genes in CS- and non-CS-associated AP. (**A**,**B**) Heatmaps show hierarchical clustering of hub gene expression in GSE194331 and GSE149607. Conditions of up-regulation and down-regulation are separated. Expression levels are normalized to the z-score. Columns of heatmaps are divided into AP and control. (**C**) Interaction graph of selected hub genes and miRNAs derived from miRTargetLink 2.0. A central component of hub genes with high node degrees is shown. (**D**) Box plots show expression of hsa-miR-21*, hsa-miR-146a, and hsa-miR-106a* between AP and control in GSE31568. Wilcoxon’s rank sum test P values are shown. * indicates the product is from the opposite arm of the miRNA precursor.

**Figure 4 biomolecules-13-00284-f004:**
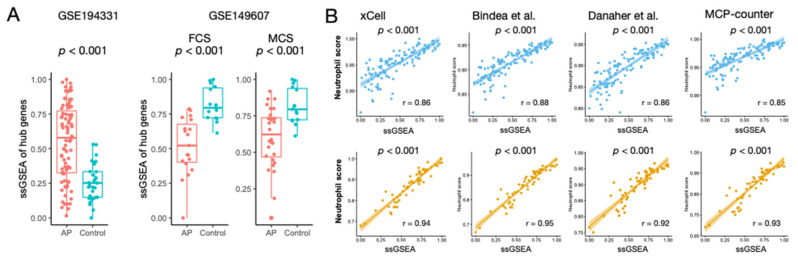
Regulatory activity of the hub genes in AP. (**A**) Box plots show differences in single-sample GSEA (ssGSEA) scores between AP and control in GSE194331 and GSE149607. Wilcoxon’s rank sum test P values are shown. (**B**) Scatter plots show Spearman’s correlation between ssGSEA scores and the estimated neutrophil infiltrations in GSE194331 and GSE149607. Gene signatures used for estimating the neutrophil infiltration are based on xCell, Bindea et al., Danaher et al., and MCP-counter. Correlation coefficients (r) are shown, and significance of correlation was obtained using Spearman’s test.

**Figure 5 biomolecules-13-00284-f005:**
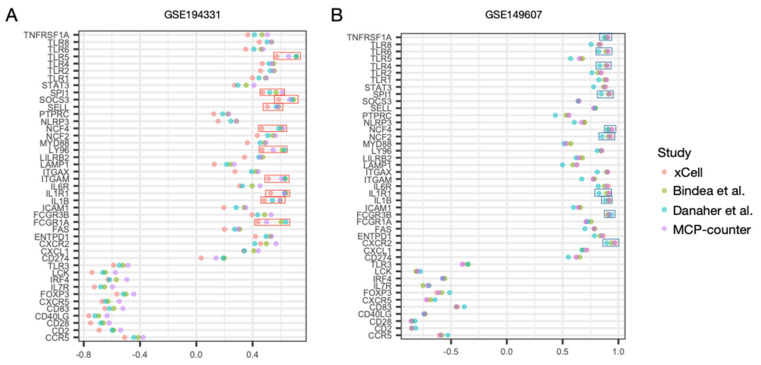
Correlation profiles of the hub genes. (**A**,**B**) Correlation coefficients calculated according to the four studies (xCell, Bindea et al., Danaher et al., and MCP-counter) between expression of the hub genes and their ssGSEA scores are plotted for GSE194331 and GSE149607. Due to higher variability between datasets, the top 10 genes with positive correlations are selected. Red boxes represent the top 10 genes with higher average correlation coefficients among the four studies in GSE194331. Blue boxes represent the top 10 genes with higher average correlation coefficients among the four studies GSE149607.

**Figure 6 biomolecules-13-00284-f006:**
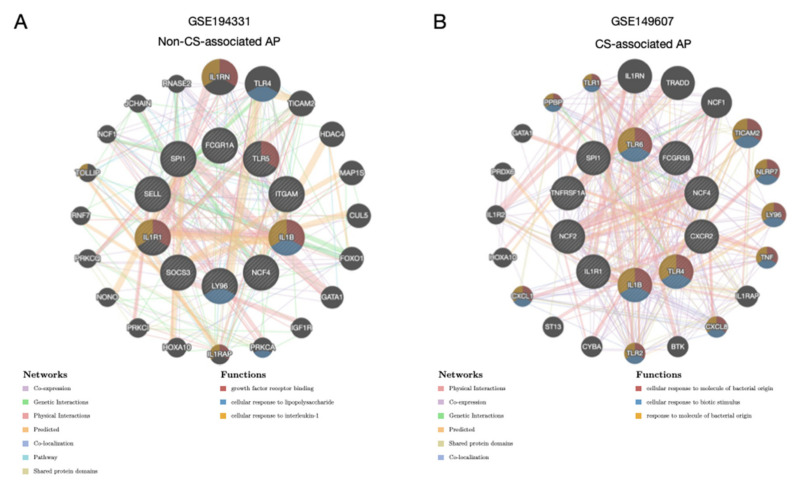
Protein–Protein networks of the hub genes. (**A**,**B**) The protein–protein interaction networks of the top 10 genes in the correlation profiles of GSE194331 and GSE149607 constructed using GeneMANIA. Interconnections between proteins are shown in terms of physical interaction, co-expression, predicted, co-localization, common pathway, genetic interaction, and shared protein domains. The top three enriched pathways are shown for the two networks.

**Figure 7 biomolecules-13-00284-f007:**
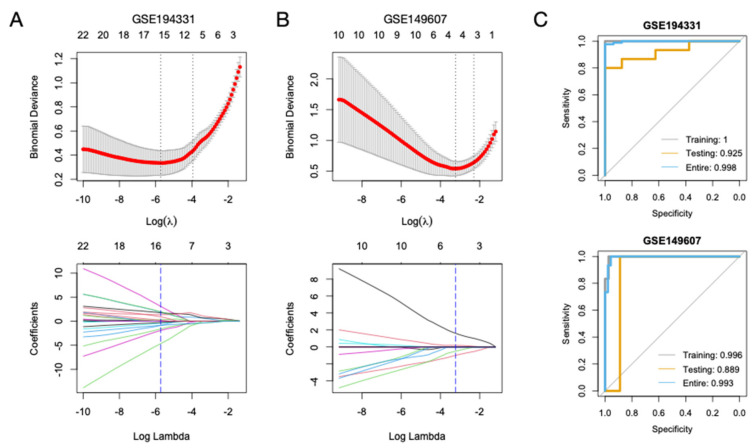
Construction of diagnostic models for AP. (**A**,**B**) Binomial deviance plot with lambda as the tuning parameter for GSE194331 and GSE149607 (upper panels). The red dots are the values of binomial deviance. The gray lines indicate the standard error (SE). The vertical dashed lines indicate the optimal values by the minimum criteria and 1-SE criteria. The least absolute shrinkage and selection operator (LASSO) coefficient profile of the selected genes is shown in the lower panels. (**C**) Receiver operating characteristic (ROC) curves of the LASSO diagnostic models in the training, testing, and entire datasets for GSE194331 (upper) and GSE14960 (lower).

**Table 1 biomolecules-13-00284-t001:** miRNAs associated with neutrophil regulation.

miRNAs	Condition	Neutrophil Source	Literature	PMID
hsa-miR-106a	Down-regulated after acute exercise	Peripheral blood	Radom-Aizik et al.	[44]
hsa-miR-21	Down-regulated after traumatic injury	Peripheral blood	Yang et al.	[45]
	Down-regulation induces apoptosis in neutrophil	Peripheral blood	Hutcheson et al.	[46]
hsa-miR-146a	Down-regulated in diffuse alveolar hemorrhage	Peripheral blood	Hsieh et al.	[47]
	Down-regulation induces neutrophil extracellular traps	Peripheral blood	Arroyo et al.	[48]
	Up-regulation represses neutrophil attraction	Peripheral blood	Meisgen et al.	[49]

## Data Availability

Data have been described throughout the text and are available under accession codes GSE194331, GSE149607, and GSE31568 from the National Center for Biotechnology Information Gene Expression Omnibus (GEO).

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
