# Peer review of "Bioinformatic Analyses of Peripheral Blood Transcriptome Identify Altered Neutrophil-Related Pathway and Different Transcriptomic Profiles for Acute Pancreatitis in Patients with and without Chylomicronemia Syndrome"

_biomolecules, 2023, doi:10.3390/biom13020284_

Round 1
Reviewer 1 Report
Review of: “Bioinformatic analyses of peripheral blood genome identify altered neutrophil-related pathway and different genomic profiles for acute pancreatitis in patients with and without chylomicronemia syndrome”
My main concern is related to the control populations. Looking at Figure 4A, we see that the AP with CS and AP without CS have fairly comparable hub gene expression, however, the controls in each dataset are completely flipped and so any Fold change will look to be in the opposite direction when comparing the DEGs between study sets. If you can address this discrepancy, then I think your manuscript can be re-evaluated. Since your results and interpretations will completely change with this reanalysis, I will stop my review at this time.
Abstract:
“Currently, no study has explored the biological differences between non-CS-associated AP and CS-associated AP.” – this gives the impression you will be doing this with your transcriptomic profiles
-you do not list the third GEO dataset you used.
Introduction:
-you do not provide evidence for why it is important to distinguish between CS and non-CS associated AP – why is it important to ID the genomic biomarkers?
-What about a review of literature related to blood-based genomic biomarkers of AP or CS?
-suggest you use transcriptomic profiles in last sentence instead of genomic profiles
Materials and methods:
-please provide citations for all the methods you outline using
-what precedence to you have to combined the different datasets the way you did? – are there studies you can cite? I am concerned about the different patient population characteristics and how that may affect your analysis. Where are each of the samples collected from and how might those differences in location affect your results? What about the case-control imbalance in some of the studies?
-Please either include your testing metrics and associated p-value cut offs in the individual sections or in section 2.11
Results:
3.1 – show the same analysis and figure (as supplement) for the with CS DEG analysis.
3.2 – “Next, DEGs from both datasets were inputted into Metascape for ORA.” – is this the overlapping set (1850) or the 2 combined DEG sets (3510+6513)?
Author Response
Response to Reviewer 1 Comments
1. My main concern is related to the control populations. Looking at Figure 4A, we see that the AP with CS and AP without CS have fairly comparable hub gene expression, however, the controls in each dataset are completely flipped and so any Fold change will look to be in the opposite direction when comparing the DEGs between study sets. If you can address this discrepancy, then I think your manuscript can be re-evaluated. Since your results and interpretations will completely change with this reanalysis, I will stop my review at this time.
Response: We appreciate the valuable comment from the reviewer.
- Since these two datasets are derived from different platforms, the generated ssGSEA values were “calculated” and “normalized” separately. Therefore, even though the ssGSEA values in AP seem comparable by looking at Figure 4A, “we can not compare them directly”. This discrepancy is a mere reflection of how distributions of control population differ from that for AP samples in both datasets.
- Based on our coding for calculating ssGSEA in R:
library(GSVA)
normalization<-function(x){
return((x-min(x))/(max(x)-min(x)))}
dat <- read.csv('/Users/arthurdai/Desktop/Gary/Datasets/GSE194331_gene_ph.csv',row.names = 1)
dat <- dat[,1:39769]
dat <- as.data.frame(t(dat))
geneLength = rowMeans(dat)
dat.conv <- convertCounts(as.matrix(dat), unit = 'tpm', geneLength = geneLength, log = FALSE, normalize = 'none')
gset_hub =c(hub_down, hub_up)
gsva_matrix0<- gsva(dat.conv, list(gset_hub), method='ssgsea',kcdf='Gaussian',abs.ranking=TRUE)
gsva_matrix0<- t(scale(t(gsva_matrix0)))
nor_gsva_matrix0 <- normalization(gsva_matrix0)
nor_gsva_matrix0 = as.data.frame(t(nor_gsva_matrix0))
colnames(nor_gsva_matrix0) = 'Hub'
We essentially scaled and normalized the dataset for visualization. Therefore the raw computed ssGSEA values were different between GSE194331 and GSE149607.
- Furthermore, as shown in Figure 3, the expression pattern of hub genes between AP with CS and AP without CS is basically “opposite”. The title in each of the heatmap indicates that the hub genes are up-regulated in one dataset, but down-regulated in another (Figure 3A-B). For example, the expression of IL1R1 is higher in AP for GSE194331 but lower in AP for GSE149607 (Figure shown below). This difference in expression pattern was then transformed into the ssGSEA values where the values are significantly higher in AP for GSE194331 but lower for GSE194331, with both conditions compared to their control groups (Figure 4A).
2. Abstract:
2.1: “Currently, no study has explored the biological differences between non-CS-associated AP and CS-associated AP.” – this gives the impression you will be doing this with your transcriptomic profiles
Response: We appreciate the valuable comment from the reviewer.
We have modified this sentence and clarified what we aimed to do for this study:
Currently, no study has explored the differences between non-CS-associated AP and CS-associated AP in terms of gene expression.
2.2: -you do not list the third GEO dataset you used.
Response: We appreciate the valuable comment from the reviewer.
We added a sentence describing our use of the third GEO dataset:
GSE31568 was used to examine the linkage between non-CS-associated AP and expression of micro RNAs (miRNAs).
3. Introduction:
3.1: -you do not provide evidence for why it is important to distinguish between CS and non-CS associated AP – why is it important to ID the genomic biomarkers?
Response: We appreciate the valuable comment from the reviewer.
According to the reviewer’s suggestion, we have added the evidence about the importance of finding biomarkers for CS-associated AP:
For both types of CS, the most daunting complication is AP, and identifying patients with CS-associated AP is essential as genetic alterations may portend therapeutic resistance, especially in FCS [1]; and identification of genomic biomarkers for this special phenotype is important for long-term management and prevention of future episodes of AP [2]. However, no study has addressed the contribution of such biomarkers to the difference between CS- and non-CS-associated AP .
3.2: -What about a review of literature related to blood-based genomic biomarkers of AP or CS?
Response: We appreciate the valuable comment from the reviewer.
We have added a paragraph about the review of literature related to blood-based genomic biomarkers.
In fact, multiple blood biomarkers have been found to be of diagnostic value for AP, such as lipase, amylase and trypsinogen [3]. These biomarkers are based on the biochemical profiling, and can sometimes be affected by other gastrointestinal diseases or disease conditions. Other indicators of AP such as those based on the metabolome, genes, cell free DNA (cfDNA) and miRNA, however, are different from the biochemical biomarkers and have not been widely investigated. These novel biomarkers are able to be integrated into machine-learning (ML) process, leading to accurate prediction of AP . Sun et al. used cfDNA methylation marker in the blood samples and constructed a prediction model for severe AP with high model performance and accuracy [4]. Zhang et al., explored the transcriptomic profiles of peripheral blood cells in AP [5]. Through ML methods, they found that S100A6, S100A9, and S100A12 were predictors of severe AP. By examining the perturbation of genomic/transcriptomic profiles in the blood samples, we can sufficiently characterize the altered biological pathways in a specific condition like CS.
3.3: -suggest you use transcriptomic profiles in last sentence instead of genomic profiles
Response: We sincerely thank the reviewer for the valuable suggestion.
We have edited this word in the revised manuscript.
4. Materials and methods:
4.1: -please provide citations for all the methods you outline using
Response: We sincerely thank the reviewer for the valuable suggestion.
We have added the citations for all the methods we used. Please check the revised manuscript. Thank you again for your suggestion.
4.2: -what precedence to you have to combined the different datasets the way you did? – are there studies you can cite? I am concerned about the different patient population characteristics and how that may affect your analysis. Where are each of the samples collected from and how might those differences in location affect your results? What about the case-control imbalance in some of the studies?
Response: We appreciate the valuable comment from the reviewer.
- In our study, we examined each dataset separately. As there were two platforms for obtaining the transcriptomic data, ie. RNA-Seq (GSE194331) and microarray (GSE149607), the data can not be merged directly because of different scale of gene expression data generated from the platforms. Additionally, the techniques for quantifying their expression are not the same. Since we attempted to identify common altered genes/pathways between two patient populations, we analyzed the data independently and pooled the results together. For example, we conducted DEG analyses separately and checked whether there was strong correlation between two datasets in terms of hub gene expression. If the correlation is strong, it means we can derive comparable results from the two datasets as the transcriptomic perturbation follows a trend for a specific condition, like AP in our study. The strong Spearman’s correlation in our study indicates that the expression of hub genes in both datasets can be compared. Therefore, we “did not combine the datasets”, but rather identified the common genes that emerged during the bioinformatic analyses and verified the suitability for subsequent analyses.
- Lim et al. calculated the Spearman’s correlation between two platforms (RNA-Seq and DNA microarray) in terms of the estimates of log 2 fold changes of gene expression values between the samples to demonstrate the comparability of the two platforms [6]. This approach was used in our study and we showed that the log 2 fold changes of gene expression values between AP and control had strong negative correlation across two platforms. This not only indicates the comparability of the two platforms in our study but also there is contrasting expression pattern for the hub genes in CS- and non-CS-associated AP.
- Different patient characteristics in this study arise from the fact that AP is contributed differently in GSE194331 and GSE149607. In GSE194331, patients with AP do not have underlying CS, whereas patients in GSE149607 had genetic predisposition toward developing AP throughout the patients’ lives. It is similar to studies that compare phenotypes between wild types and mutants. The purpose of such studies is to address the change of phenotypes due to shifts in genotypes and their relationships [7].
- Zhang et al. used the same GEO dataset in their study (GSE194331) and identified the same enriched biological pathway (Neutrophil degranulation) by the enrichment analysis [5]. In our study, we applied similar methodologies, and therefore we think our results are feasible and could be referenced to other studies.
- Each sample is obtained from blood. This can be accessed from any location where blood sample is available for drawing. It does not affect our analyses as the blood samples reflect the general status of our body system.
- In the case of large genomic analysis such as GWAS, case-control ratios are often unbalanced (case:control=1:10) or extremely unbalanced (case:control<1:100) when the prevalence of a condition or a disease is low [8]. This can pose tremendous challenge and potentially increase the type I error rate during association study. However, in our study, even though the datasets were imbalanced, the case-control ratios are 1:0.37 in GSE194331 and 1:0.32 in GSE149607, respectively. Moreover, in our study, the datasets were mainly used for differential gene expression analysis. By using DESeq2, average gene expression is compared to derive fold change. Thus, slight case-control imbalance might not affect the DEGs. However, according to the designer’s recommendation (https://support.bioconductor.org/p/9142704/), there is nothing to do if there is extreme imbalance, but this situation is not present in our study.
- Additionally, the prediction model, without data augmentation and balancing, showed high AUC in training and testing datasets, suggesting the presence of case-control imbalance did not dent the model performance. However, this should still be validated by external independent datasets.
4.3: -Please either include your testing metrics and associated p-value cut offs in the individual sections or in section 2.11
Response: We sincerely thank the reviewer for the valuable suggestion.
2.2 Differential expression analysis
For GSE194331, differentially expressed genes (DEGs) were identified in patients with AP with varying severities and healthy controls through a linear regression model using the R/Bioconductor limma package [9]. Significance was defined as a false discovery rate (FDR)-adjusted P-value (adj.P) < 0.05.
2.3 Gene set enrichment analysis (GSEA)
A ranked list was generated according to the log fold change (logFC) for DEGs between all AP cases and healthy controls. GSEA was performed using the GSEA function of the R/Bioconductor clusterProfiler package [10]. The enrichment score (ES) was derived by calculating the weighted Kolmogorov-Smirnov statistic to a running sum of the ranked list. ES was further normalized to account for the size of each gene set. FDR’s less than 0.05 were considered statistically significant.
2.4 Over-representation analysis (ORA)
To identify the pathways involved in AP, common DEGs between GSE194331 and GSE149607 were inputted into Metascape (https://metascape.org/), a website designed to provide comprehensive gene list annotation and pathway enrichment analysis [11]. During enrichment analysis, the input DEGs were compared to thousands of gene sets curated from various sources (KEGG Pathway, GO Biological Processes, Reactome Gene Sets, Canonical Pathways, CORUM, and WikiPathways) to identify their involvement in specific biological processes. Enriched terms were defined by hypergeometric test and Benjamini-Hochberg P correction algorithm. P < 0.05 was considered significant [11].
2.9 Prediction of associated genes and biological pathways
We hypothesized that the expression profiles of hub genes would be different between patients with and without CS. To confirm this, we first calculated Spearman’s correlation coefficients between hub genes and specific immune cell types of interest. The correlation coefficients were then averaged across four studies (xCell, MCP-counter, Bindea et al., and Danaher et al.) and ranked. We then extracted the top ten genes with positive correlation coefficients and used GeneMANIA to construct new PPI networks [12]. GeneMANIA is an online server where interconnections between proteins can be explored in terms of physical interactions, co-expression, predicted, and co-localization. Pathway analysis based on the DAVID website was performed to identify the functional enrichment of the top ten genes and interacting genes. Enrichment is measured by hypergeometric P value and P < 0.05 was considered significant.
5. Results:
5.1 – show the same analysis and figure (as supplement) for the with CS DEG analysis.
Response: We sincerely thank the reviewer for the valuable suggestion.
We have conducted the same analyses for AP with CS and put the results in the supplement.
Please kindly check the revised files and manuscript.
In GSE149607, there was clustering of control samples that was separate from FCS and MCS (Supplementary Figure 1A). The most prominent DEGs were identified in the control samples, which were MS4A2, CPA3 and LRG1 (Supplementary Figure 1B). GSEA results showed 3819 enriched GO terms and GO:0042581 was also significantly enriched, but with the opposite regulatory pattern as compared with that in GSE194331 (Normalized ES:–2.68, adjusted P < 0.001, Supplementary Figure 1C-D). Based on the findings derived from the blood samples, these results suggest that neutrophil-related biological processes play vital roles in AP and the regulatory activity might be different between CS-associated and non-CS-associated AP.
5.2 – “Next, DEGs from both datasets were inputted into Metascape for ORA.” – is this the overlapping set (1850) or the 2 combined DEG sets (3510+6513)?
Response: We sincerely thank the reviewer for the kind suggestion.
The shared DEGs were used. We have edited the original sentence to clarify this.
Next, the shared DEGs were inputted into Metascape for ORA.
Thank you again for your important comment. References
1. Goldberg, R.B.; Chait, A. A Comprehensive Update on the Chylomicronemia Syndrome. Front Endocrinol (Lausanne) 2020, 11, 593931, doi:10.3389/fendo.2020.593931.
2. Yang, A.L.; McNabb-Baltar, J. Hypertriglyceridemia and acute pancreatitis. Pancreatology 2020, 20, 795-800, doi:10.1016/j.pan.2020.06.005.
3. Meher, S.; Mishra, T.S.; Sasmal, P.K.; Rath, S.; Sharma, R.; Rout, B.; Sahu, M.K. Role of Biomarkers in Diagnosis and Prognostic Evaluation of Acute Pancreatitis. J Biomark 2015, 2015, 519534, doi:10.1155/2015/519534.
4. Sun, H.W.; Dai, S.J.; Kong, H.R.; Fan, J.X.; Yang, F.Y.; Dai, J.Q.; Jin, Y.P.; Yu, G.Z.; Chen, B.C.; Shi, K.Q. Accurate prediction of acute pancreatitis severity based on genome-wide cell free DNA methylation profiles. Clin Epigenetics 2021, 13, 223, doi:10.1186/s13148-021-01217-z.
5. Zhang, D.; Wang, M.; Zhang, Y.; Xia, C.; Peng, L.; Li, K.; Yin, H.; Li, S.; Yang, X.; Su, X., et al. Novel insight on marker genes and pathogenic peripheral neutrophil subtypes in acute pancreatitis. Front Immunol 2022, 13, 964622, doi:10.3389/fimmu.2022.964622.
6. Lim, S.B.; Tan, S.J.; Lim, W.T.; Lim, C.T. An extracellular matrix-related prognostic and predictive indicator for early-stage non-small cell lung cancer. Nat Commun 2017, 8, 1734, doi:10.1038/s41467-017-01430-6.
7. Orgogozo, V.; Morizot, B.; Martin, A. The differential view of genotype-phenotype relationships. Front Genet 2015, 6, 179, doi:10.3389/fgene.2015.00179.
8. Zhou, W.; Nielsen, J.B.; Fritsche, L.G.; Dey, R.; Gabrielsen, M.E.; Wolford, B.N.; LeFaive, J.; VandeHaar, P.; Gagliano, S.A.; Gifford, A., et al. Efficiently controlling for case-control imbalance and sample relatedness in large-scale genetic association studies. Nat Genet 2018, 50, 1335-1341, doi:10.1038/s41588-018-0184-y.
9. Ritchie, M.E.; Phipson, B.; Wu, D.; Hu, Y.; Law, C.W.; Shi, W.; Smyth, G.K. limma powers differential expression analyses for RNA-sequencing and microarray studies. Nucleic Acids Res 2015, 43, e47, doi:10.1093/nar/gkv007.
10. Yu, G.; Wang, L.G.; Han, Y.; He, Q.Y. clusterProfiler: an R package for comparing biological themes among gene clusters. OMICS 2012, 16, 284-287, doi:10.1089/omi.2011.0118.
11. Zhou, Y.; Zhou, B.; Pache, L.; Chang, M.; Khodabakhshi, A.H.; Tanaseichuk, O.; Benner, C.; Chanda, S.K. Metascape provides a biologist-oriented resource for the analysis of systems-level datasets. Nat Commun 2019, 10, 1523, doi:10.1038/s41467-019-09234-6.
12. Warde-Farley, D.; Donaldson, S.L.; Comes, O.; Zuberi, K.; Badrawi, R.; Chao, P.; Franz, M.; Grouios, C.; Kazi, F.; Lopes, C.T., et al. The GeneMANIA prediction server: biological network integration for gene prioritization and predicting gene function. Nucleic Acids Res 2010, 38, W214-220, doi:10.1093/nar/gkq537.
Reviewer 2 Report
In this manuscript, Liu and Dai reported the bioinformatic analysis of the transcriptome profiles of acute pancreatitis in patients with and without chylomicronemia syndrome. With the published datasets from GEO, they identified shared DEGs and found neutrophil degranulation pathways were significantly enriched in both non-CS-associated and CS-associated AP. The authors also identified 43 common hub genes in patients with and without CS, which also revealed high diagnostic values for AP. The hub genes derived ssGSEA scores showed patients without CS had significantly higher ssGSEA scores.
Overall, this manuscript utilized the published datasets, designed proper analysis strategies, and demonstrated the conclusions appropriately. Before the final publication, there are several minor issues should be addressed.
Minor issues:
1. Regarding the diagnostic values, the authors should discuss the limitation too. The gene expression analysis was conducted by comparing AP samples with healthy controls, the results were limited to the tell the difference in AP samples. However, if a patient has other kinds of injury or inflammation, their peripheral blood sample may have a similar gene expression profile. So, this diagnostic value must be combined with other evidence to link to AP. Besides this, for diagnostic purposes, a rapid turnaround time is important for AP. However, to generate the gene expression profile, how long it will take and whether it is fast enough to support clinical diagnosis should be discussed.
2. The genome/genomic in the title should change to transcriptome/transcriptomic since only gene expression status was considered.
3. In Section 3.1 and Figure 1C, the XIST did not reach the significant cutoff, but why was it highlighted and mentioned as the most prominent DEG?
4. In Figure 1A, the color of Mild AP and Severe AP are difficult to distinguish. The author should consider changing the color. And what is the dotted line?
5. In Figure 1F, this is a bad way to illustrate the expression level of multiple genes. The author should change to use a heatmap, or box plot but group by genes (each gene has four boxes showing the expression level of control/Mild/Moderately/Severe AP)
6. In figure 2B, use the term name/description instead of using the term IDs in the y-axis labels.
7. In Figure 3 legend, the authors didn't mention what does the "*" mean.
8. The authors should pay attention to the consistency of terms used in the manuscript. Like the italic font of gene names (not all the places using the right form, especially in figures), "adjusted P-value", "P-value", LogFC/Log2FC, etc.
Author Response
Response to Reviewer 2 Comments
In this manuscript, Liu and Dai reported the bioinformatic analysis of the transcriptome profiles of acute pancreatitis in patients with and without chylomicronemia syndrome. With the published datasets from GEO, they identified shared DEGs and found neutrophil degranulation pathways were significantly enriched in both non-CS-associated and CS-associated AP. The authors also identified 43 common hub genes in patients with and without CS, which also revealed high diagnostic values for AP. The hub genes derived ssGSEA scores showed patients without CS had significantly higher ssGSEA scores.
Overall, this manuscript utilized the published datasets, designed proper analysis strategies, and demonstrated the conclusions appropriately. Before the final publication, there are several minor issues should be addressed.
Minor issues:
- Regarding the diagnostic values, the authors should discuss the limitation too. The gene expression analysis was conducted by comparing AP samples with healthy controls, the results were limited to the tell the difference in AP samples. However, if a patient has other kinds of injury or inflammation, their peripheral blood sample may have a similar gene expression profile. So, this diagnostic value must be combined with other evidence to link to AP. Besides this, for diagnostic purposes, a rapid turnaround time is important for AP. However, to generate the gene expression profile, how long it will take and whether it is fast enough to support clinical diagnosis should be discussed.
Response: We appreciate the valuable comment from the reviewer.
We have addressed the limitations further in the “Discussion”:
Our study had several limitations. First, we analyzed only three datasets. Therefore, the power to obtain conclusive results may be limited. Second, there was a lack of data from the pancreatic tissue of the same study population. Therefore, we could not evaluate the difference in neutrophil infiltration between blood and tissue samples. Third, information on the TG levels in patients with CS is unknown. Thus, the relationship between TG level and estimated neutrophil infiltration could not be elucidated. Due to the rarity of FCS and MCS, we were unable to find suitable patients in our hospital to confirm the results. Despite these limitations, our study may provide insights into HTGP and other relevant disorders. Furthermore, as we derived the results only from patients with AP, other inflammatory conditions may have similar gene expression profiles when neutrophil plays a dominant role. Other clinical features related to diagnosing AP should be incorporated to improve the diagnostic yield. Lastly, the time for generating results from DNA microarray or RNA-Seq might be approximately one to two days [1]. It could be lengthy when immediate management of AP is necessary. We should still follow the current management guideline to prevent mortality, while at the same time wait for the results that address the patient’s underlying causes of AP. This approach could help clinicians spend time grappling with the pathogenesis and managing this disease in a long-term.
2. The genome/genomic in the title should change to transcriptome/transcriptomic since only gene expression status was considered.
Response: We appreciate the valuable comment from the reviewer.
We have changed the title of our manuscript. Thank you again for your crucial suggestion.
3. In Section 3.1 and Figure 1C, the XIST did not reach the significant cutoff, but why was it highlighted and mentioned as the most prominent DEG?
Response: We appreciate the valuable comment from the reviewer.
We have re-generated the volcano plot and removed XIST from the plot. In the revised manuscript, we also removed it from the most prominent DEGs list.
Thank you again for your crucial suggestion.
4. In Figure 1A, the color of Mild AP and Severe AP are difficult to distinguish. The author should consider changing the color. And what is the dotted line?
Response: We appreciate the valuable comment from the reviewer.
We have changed the color the make it clear. Thank you again for your important and valuable advice.
5. In Figure 1F, this is a bad way to illustrate the expression level of multiple genes. The author should change to use a heatmap, or box plot but group by genes (each gene has four boxes showing the expression level of control/Mild/Moderately/Severe AP)
Response: We appreciate the valuable comment from the reviewer.
We have removed the original box plot and generated a new heatmap for this.
Thank you again for your crucial suggestion.
6. In figure 2B, use the term name/description instead of using the term IDs in the y-axis labels.
Response: We appreciate the valuable comment from the reviewer.
We have added the term description in addition to the term ID:
7. In Figure 3 legend, the authors didn't mention what does the "*" mean.
Response: We appreciate the valuable comment from the reviewer.
We have added the description of “*” sign below Figure 3:
* indicates the product is from the opposite arm of the miRNA precursor.
Thank you again for your valuable suggestion.
8. The authors should pay attention to the consistency of terms used in the manuscript. Like the italic font of gene names (not all the places using the right form, especially in figures), "adjusted P-value", "P-value", LogFC/Log2FC, etc.
Response: We appreciate the valuable comment from the reviewer.
We have edited some of the inconsistent terms or names in our revised manuscript. Thank you again for your crucial advice.
References
1. Narrandes, S.; Xu, W. Gene Expression Detection Assay for Cancer Clinical Use. J Cancer 2018, 9, 2249-2265, doi:10.7150/jca.24744.
